# Size-dependent loss of aboveground animals differentially affects grassland ecosystem coupling and functions

A.C. Risch [1], R. Ochoa-Hueso[2], W.H. van der Putten [3,4], J.K. Bump [5,6], M.D. Busse[7], B. Frey [8] D.J. Gwiazdowicz[9], D.S. Page-Dumroese[10], M.L. Vandegehuchte [1,11], S. Zimmermann [8] & M. Schütz[1]

Increasing evidence suggests that community-level responses to human-induced biodiversity loss start with a decrease of interactions among communities and between them and their abiotic environment. The structural and functional consequences of such interaction losses are poorly understood and have rarely been tested in real-world systems. Here, we analysed how 5 years of progressive, size-selective exclusion of large, medium, and small vertebrates and invertebrates—a realistic scenario of human-induced defaunation—impacts the strength of relationships between above- and belowground communities and their abiotic environment (hereafter ecosystem coupling) and how this relates to ecosystem functionality in grasslands. Exclusion of all vertebrates results in the greatest level of ecosystem coupling, while the additional loss of invertebrates leads to poorly coupled ecosystems. Consumer-driven changes in ecosystem functionality are positively related to changes in ecosystem coupling. Our results highlight the importance of invertebrate communities for maintaining ecological coupling and functioning in an increasingly defaunated world.

[1] Community Ecology, Swiss Federal Institute for Forest, Snow and Landscape Research WSL, Zuercherstrasse 111, 8903 Birmensdorf, Switzerland.
[2] Department of Ecology, Autonomous University of Madrid, 2, Darwin Street, 28049 Madrid, Spain. [3] Netherlands Institute of Ecology, Droevendaalsesteeg 10, 6708 PB Wageningen, The Netherlands. [4] Laboratory of Nematology, Wageningen University, P.O. Box 81236700 ES Wageningen, The Netherlands. [5] School of Forest Resources and Environmental Sciences, Michigan Technological University, 1400 Townsend Drive, Houghton, MI 49931, USA. [6] Department of Fisheries, Wildlife, & Conservation Biology, University of Minnesota – Twin Cities, 135 Skok Hall, 2003 Upper Buford Circle, St. Paul, MN 55108, USA. [7] Pacific Southwest Research Station, USDA Forest Service, 1731 Research Park, Davis, CA 95618, USA. [8] Forest Soils and Biogeochemistry, Swiss Federal Institute for Forest, Snow and Landscape Research WSL, Zuercherstrasse 111, 8903 Birmensdorf, Switzerland. [9] Faculty of Forestry, Poznan University of Life Sciences, Wojska Polskiego 71c, 60 625 Poznań, Poland. [10] Rocky Mountain Research Station, USDA Forest Service, 1221 South Main St, Moscow, ID 83843, USA. [11] Terrestrial Ecology Unit, Department of Biology, Ghent University, K. L. Ledeganckstraat 35, 9000 Ghent, Belgium. These authors contributed equally: A.C. Risch, R. Ochoa-Hueso. Correspondence and requests for materials should be addressed to A.C.R. (email: anita.risch@wsl.ch) or to R.O.-H. (email: rochoahueso@gmail.com)

Rapid human-induced decline in global biodiversity across all trophic levels[1–6] reduces the ability of ecosystems to maintain key ecosystem functions[2,3,7]. As species engage in numerous and often-hidden interactions with other species and their physicochemical environment[8], a loss of ecological interaction may have far-reaching consequences for the functioning of ecosystems. In fact, it has been proposed that ecological interactions may disappear well before species do[8]. An approach to investigate the consequences of losing interactions between multispecies communities and their environment is to analyse the degree of ecosystem coupling[9], defined as the overall strength of correlation-based associations between above- and belowground plant, animal and microbial communities, and of these communities with their surrounding physicochemical environment[9]. Visually and analytically, ecosystem coupling can be represented as a network in which individual species are substituted with multispecies communities (e.g., microbes, plants and nematodes). Under undisturbed conditions we would expect that the communities are more strongly connected with one another and their abiotic environment than under disturbed conditions[10–12]. The connections can result from both direct and indirect, positive or negative interactions, depending on the communities and the dominant mechanisms involved (e.g., predation, parasitism and competition)[13]. A greater number of significant correlations, regardless of their sign, represents greater ecosystem coupling.

Extirpation of vertebrates and invertebrates from ecosystems, i.e., defaunation[1,4,5], alters the structure, composition and strength of correlation networks within and between trophic levels[13], potentially weakening interactions such as those between the aboveground invertebrate and belowground decomposer communities[14], or between the soil arthropod community and their physicochemical environment[4]. However, only a few studies have investigated such responses in real-world ecosystems and under realistic scenarios of animal biodiversity loss[1,4,15,16]. Thus, relatively little is known about the consequences of defaunation for ecosystem coupling. Moreover, defaunation often operates in a size-selective, non-random manner, with larger species disappearing before smaller-sized species[4,17]. Although vertebrates have been progressively excluded in the size-selective order in which they are expected to disappear[18–21], we are not aware of any study that has considered how the progressive, size-based loss of different vertebrates and invertebrates affects the strength of ecosystem coupling. In addition, it is not known how defaunation-induced changes in ecosystem coupling may affect ecosystem functioning. Stronger interactions among communities and between communities and their environment should lead to greater ecosystem functionality due to more efficient transfer of nutrients and energy through the system[22,23], which should result in a greater ability to withstand environmental stress[7].

We carried out a 5-year field experiment in subalpine grasslands in which we progressively excluded large, medium and small mammalian vertebrates, and, ultimately, all aboveground-dwelling invertebrates with size-selective fences[24,25] (Fig. 1 and Supplementary Table 1). Large mammals are often assumed to drive trophic interactions via top–down effects and impact other communities and abiotic properties via predation, grazing,

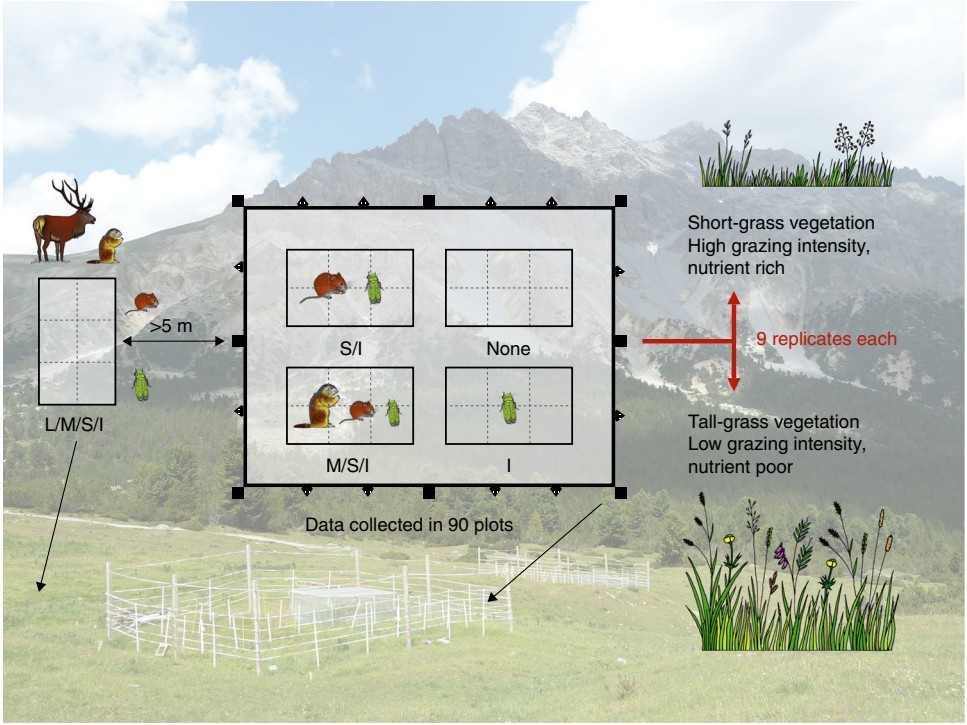

**Fig. 1** Size-selective fences to progressively exclude vertebrates and invertebrates. Nine exclosure setups were established in each of the two different vegetation types; short-grass and tall-grass vegetation. The 2 × 3 m plots inside the main 7 × 9 m fence were randomly assigned an exclusion treatment in each of our 18 size-selective exclosure setups. Data were collected on a total of 90 plots, from which we calculated ecosystem coupling and multifunctionality. The photo in the background shows two of our 18 exclosure setups in the field. We use the following abbreviations to describe our treatments throughout the manuscript: L/M/S/I = Large mammals, medium mammals, small mammals, and invertebrates have access, M/S/I = Medium mammals, small mammals, and invertebrates have access, S/I = Small mammals and invertebrates have access, I = Invertebrates have access, None = No animals have access. L/M/S/I plots (reference plots) are located outside of the fences as indicated by the arrow. Martijn L. Vandegehuchte created the animal and vegetation images for use in this paper. The animal images were adapted from Vandegehuchte et al.[40]. Anita C. Risch took the background photo and created the design image, Pablo Hueso composed the Figure

resource competition or facilitation[26–29]. We therefore predicted that the loss of large animals would reduce ecosystem coupling more than a subsequent loss of smaller animals. Expecting that stronger links between communities and between communities and their environment result in a more efficient flow of energy and matter through the ecosystem[22,23], we also predicted a positive relationship between ecosystem coupling and functionality.

The field experiment encompassed 18 size-selective exclosure setups distributed across two clearly differentiated vegetation types in the Swiss National Park (SNP; nine in short-grass, nine in tall-grass vegetation; Fig. 1). The vegetation types differ in grazing intensity and productivity due to historical differences in land use prior to the park's foundation (1914)[24,30] and represent a long-term trajectory of changes in grazing regimes[30]. Each exclosure setup contained four treatment plots that progressively excluded large, medium and small-sized mammals and aboveground-dwelling invertebrates[24] (Fig. 1). Next to each exclosure (>5 m away) a reference plot provided access to all animals. The grasslands under study can be considered natural, as humans have not directly disturbed the ecological processes within the SNP since 1914 (protection status IUCN category Ia[31]).

We calculated how our treatments affected ecosystem coupling using correlations between multivariate components [principal component analyses (PCA) axes scores] that represent the structure and composition of ecological communities as well as correlations between these components and the physicochemical environment[9]. As such, ecosystem coupling is a measure of how communities (not species) respond to alterations of the ecosystem[9] as happens through eutrophication, climate warming or species loss[1,4,32,33]. Ecosystem coupling can then be compared against a null model where the average correlation strength (absolute value of pairwise Spearman rank correlations) does not differ from random[22,34,35]. The greater the mean of absolute correlations, the more tightly coupled the ecosystem is. We considered both biotic–biotic and abiotic–biotic correlations, and used a total of 14 abiotic and biotic ecosystem constituents: soil bulk density, pH, carbon (C) content, moisture (Supplementary Table 2), as well as the first two PCA axes scores of the community composition of soil microorganisms, nematodes, arthropods, aboveground-dwelling invertebrates and plants. We assessed how the progressive animal exclusions affected overall ecosystem coupling, i.e., the average correlation strength, including both above–belowground biotic–biotic and abiotic–biotic interactions, as well as the coupling between above–belowground biotic–biotic and abiotic–biotic components separately. We also calculated overall, biotic–biotic and abiotic–biotic ecosystem coupling based on belowground constituents only to evaluate the potential effects of our treatments on belowground coupling (i.e., soil networks)[22]. Note that abiotic–abiotic correlations were not included in our analyses as the focus of this study was on interactions involving communities.

We considered six ecosystem functions and process rates:[36] soil net nitrogen (N) mineralisation, soil respiration, plant tissue N content, plant species richness, root biomass, and microbial biomass carbon (Supplementary Table 2), and calculated ecosystem multifunctionality based on the multiple threshold approach[36,37]. We then assessed how our ecosystem coupling measures were related to ecosystem functions as well as to ecosystem multifunctionality.

Excluding all mammals, results in the greatest level of ecosystem coupling, while the additional loss of invertebrates leads to poorly coupled ecosystems. Changes in ecosystem functionality are positively related to ecosystem coupling. Our findings highlight the importance of invertebrate communities for maintaining ecosystems in an increasingly defaunated world.

## Results and Discussion

**Defaunation and ecosystem coupling**. Our size-selective exclusions of aboveground vertebrates and invertebrates changed the overall degree of ecosystem coupling with comparable trends across the two vegetation types (Figs. 2a, 3, and Supplementary Fig 1a). Progressive exclusion led to greater ecosystem coupling, reaching highest values when all mammals were excluded, although coupling values dropped again when no animals (mammals + invertebrates) were present (Fig. 2a and Supplementary Table 3). In the heavily grazed short-grass vegetation[24,30], ecosystem coupling was, however, only significantly different from the null model when all mammals were excluded (Supplementary Fig 1a and Supplementary Table 3). In addition, we found the greatest biotic–biotic coupling in this vegetation type when all animals were present (Supplementary Fig 1b and Supplementary Table 3). This suggests strong top–down effects of large ungulates on other communities in the short-grass vegetation, in agreement with our hypothesis and previous studies[30,38–40]. This tight biotic–biotic coupling (Supplementary Fig 1b) was countered by a low abiotic–biotic coupling (Supplementary Fig 1c) in the situation where all animals, including large mammals, were present (Supplementary Fig 1a and Supplementary Table 3). In the tall-grass vegetation, which is under comparatively lower grazing pressure[24,30], all treatments where animals had access were significantly coupled, although only marginally when all animals were present (permutation-based, $p = 0.07$; Supplementary Fig 1a and Supplementary Table 3).

Interestingly, invertebrates alone, i.e., in the absence of all mammals, were able to maintain or even increase ecosystem coupling regardless of vegetation type (Fig. 2a, Supplementary Fig 1a and Supplementary Table 3). However, in the short-grass vegetation, coupling within the invertebrates-only treatment was due to stronger abiotic–biotic interactions (Supplementary Fig 1c), while in the tall-grass vegetation biotic–biotic interactions were the main drivers of coupling in this treatment (Supplementary Fig 1b and Supplementary Table 3). Thus, shifts in the strength of biotic–biotic and abiotic–biotic coupling due to progressive animal exclusion had a profound impact on the way communities were connected with one another and with their abiotic environment, in particular when all mammals were missing, in which case overall coupling was highest (Fig. 2, Supplementary Fig 1 and Supplementary Table 3). When only considering belowground interactions, i.e., between soil biotic and abiotic constituents, we found similar patterns compared to when all above–belowground interactions were included, although coupling values did not differ from random when large and medium-sized mammals were present (Fig. 2d–f and Supplementary Fig. 1d–f). These findings indicate that belowground interactions are an important component of overall ecosystem coupling. The greater variation in coupling for soil networks and the less significant results compared to the overall networks is likely due to the small number of belowground-only links compared to the overall number of interactions and not a reflection of the underlying biology of the system.

**Defaunation and ecosystem functioning**. Together with treatment responses in ecosystem coupling, four of our six individual ecosystem functions responded to the experimental exclusion treatments (Fig. 4a–f). Average ecosystem multifunctionality calculated from all thresholds between 30 and 80% was only slightly affected by the treatments ($F_{4,64} = 2.22$, permutation-

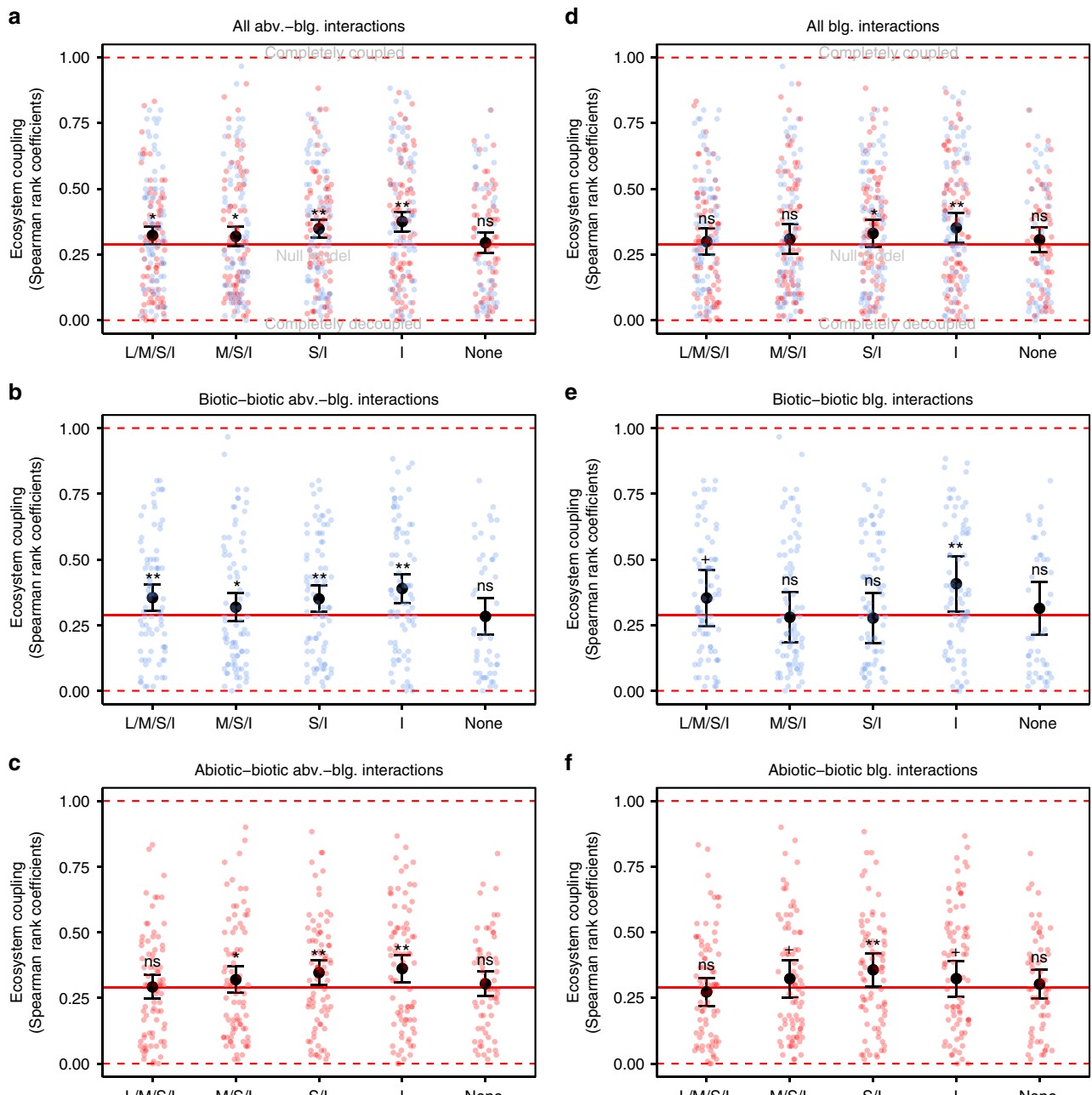

**Fig. 2** Effect of progressive animal exclusion on ecosystem coupling. Ecosystem coupling (absolute values of Spearman's rho) calculated based on **a** all interactions, **b** biotic–biotic interactions, **c** abiotic–biotic interactions involving above- and belowground constituents, and **d** all interactions, **e** biotic–biotic interactions, **f** abiotic–biotic soil interactions involving belowground constituents only. Each treatment includes interactions calculated for both vegetation types (i.e., short-grass and tall-grass) separately. All above-belowground interactions: $n = 160$ ($n = 112$ in the case of the 'None' treatment). Biotic–biotic above-belowground interactions: $n = 80$ ($n = 48$ in the case of the 'None' treatment). Abiotic–biotic above-belowground interactions: $n = 80$ ($n = 64$ in the case of the 'None' treatment). All belowground interactions: $n = 72$. Biotic–biotic belowground interactions: $n = 24$. Abiotic–biotic belowground interactions: $n = 48$. Red line: null model below which average correlation happens by chance. Red dashed lines: greatest/smallest coupling values possible. Error bars: 95% confidence interval of the mean. Notations above the confidence intervals (ns, +, *, **) indicate statistical, permutation-based differences from the null model: ns not significant, +$p$ value = 0.05–0.1, *$p$ value = 0.05–0.01, **$p$ value <0.01. All $p$ values can be found in Supplementary Table 3. Background points: individual interactions between biotic–biotic (blue) and abiotic–biotic (red) constituents. L/M/S/I = Large/medium/small mammals, and invertebrates have access, M/S/I = Medium/small mammals, and invertebrates have access, S/I = Small mammals and invertebrates have access, I = Invertebrates have access, None = No animals have access (see Fig. 1). Abv. = aboveground, blg. = belowground

based $p = 0.08$, Fig. 4g). However, significant treatment differences in multifunctionality were detected at the 50% threshold (linear mixed-effects model: $F_{4,64} = 4.51$, $p = 0.003$). This is the threshold at which the ecosystem coupling–multifunctionality relationship was tightest. Somewhat lower multifunctionality was

found when large mammals were still present or when all animals were absent, while the presence of medium- and small-sized mammals and/or invertebrates resulted in greater multifunctionality. While the decline of multifunctionality with the additional loss of invertebrates warns about the importance of

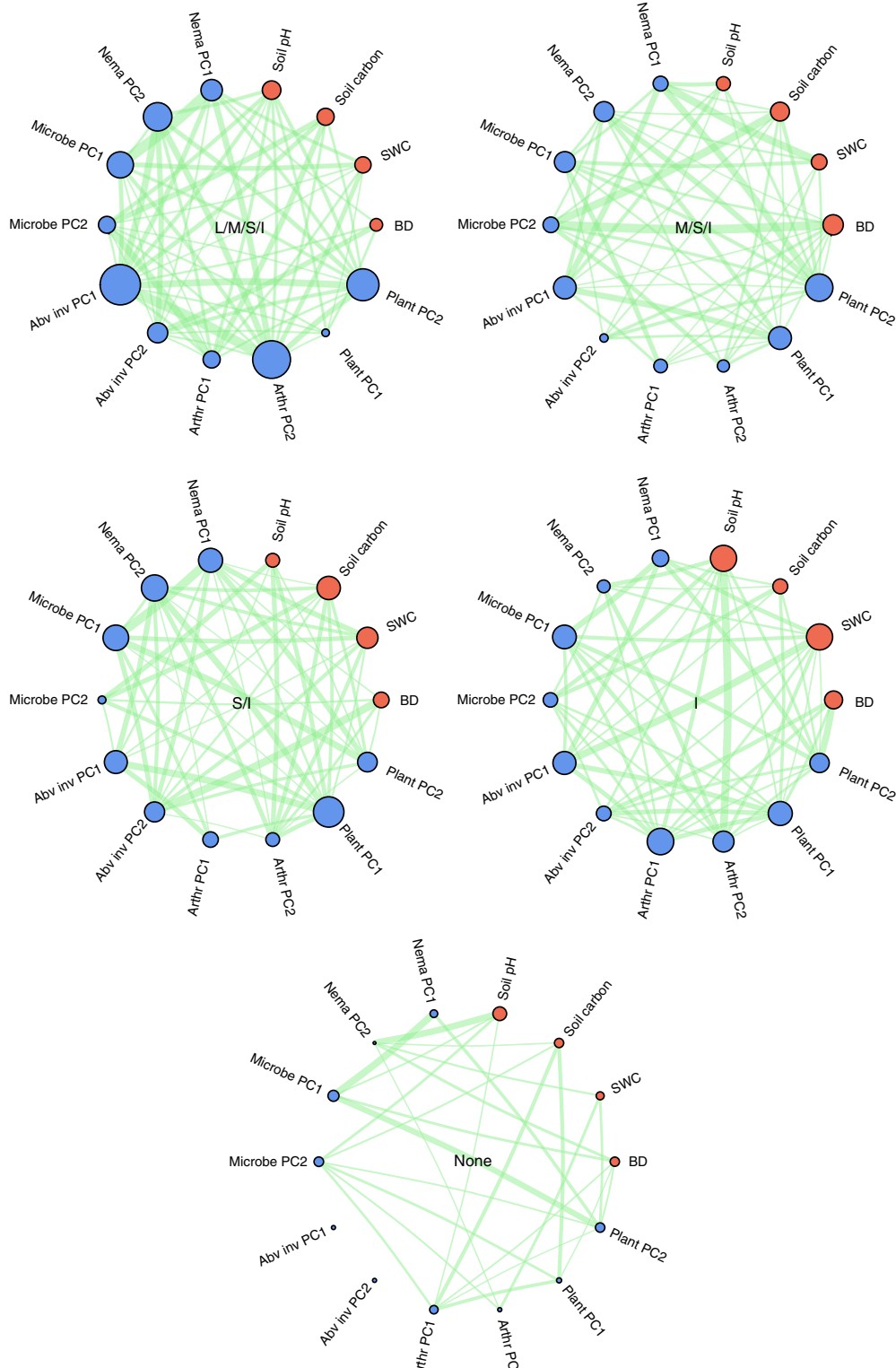

**Fig. 3** Effect of progressive animal exclusion on correlations between constituents. Network graphs with absolute Spearman rank correlations between constituents for each fence type and both vegetation types together. Green lines: pairwise interactions that exceed the expected strength from the null model. Green line width: proportional strength of the association between two constituents. Blue circles: biotic constituents, red circles: abiotic constituents, circle size: proportional average strength of pairwise interactions involving a particular constituent. L/M/S/I = Large/medium/small mammals, and invertebrates have access, M/S/I = Medium/small mammals, and invertebrates have access, S/I = Small mammals and invertebrates have access, I = Invertebrates have access, None = No animals have access (see Fig. 1). PC1, PC2 = principal component 1 and 2, Nema = nematodes, Micro = microbes, Abv = aboveground, inv = invertebrates, Arthr = soil arthropods, BD = soil bulk density, SWC = soil moisture

this group for ecosystem functionality in a progressively defaunated world[1,5,10,41,42], the slightly lower average multifunctionality found when large mammals were present should not be interpreted as a sign that their presence is negative for ecosystem functioning. Instead, this lower-average functionality might reflect the strong top–down control that these animals can have in the absence of large predators (e.g., through biomass consumption and physical disturbance)[1,41], as found in our

system where wolves, bears and lynx were locally extinct by the late nineteenth century and remain functionally absent.

**Changes in ecosystem coupling and functioning are related.** More tightly coupled ecosystems may support a wider range of functions, which could be associated with a greater efficiency in the use of resources and the processing of organic matter[22,23]. Indeed, soil net N mineralisation and soil microbial biomass C

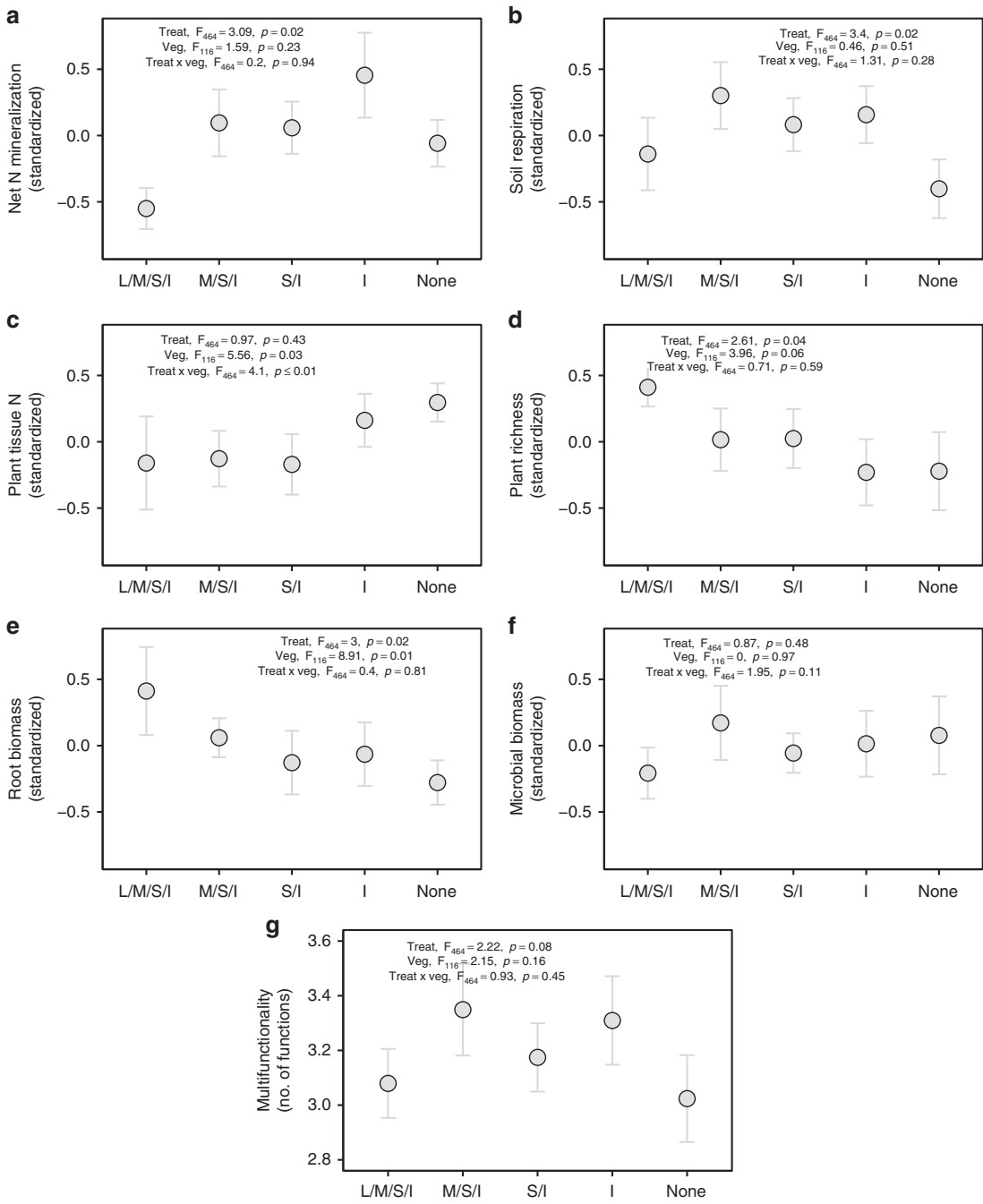

**Fig. 4** Effect of progressive animal exclusion on ecosystem functions and multifunctionality. **a** Soil net nitrogen (N) mineralisation, **b** soil respiration, **c** plant tissue nitrogen (N) content, **d** plant species richness, **e** root biomass, **f** microbial biomass carbon (MBC), **g** multifunctionality. Individual functions are standardised and were used to calculate ecosystem multifunctionality. Statistics refer to the results of linear mixed-effects models, with treatment (Treat) and vegetation type (Veg) as fixed factors and fence as a random factor. Mean values and standard errors of all individual functions across treatments can be found in Supplementary Table 2. $n = 18$. L/M/S/I = Large/medium/small mammals, and invertebrates have access, M/S/I = Medium/small mammals, and invertebrates have access, S/I = Small mammals and invertebrates have access, I = Invertebrates have access, None = No animals have access (see Fig. 1)

were related to several of our ecosystem coupling measures (Supplementary Table 4). We found that ecosystem coupling was positively related to ecosystem multifunctionality at thresholds between 36 and 65% when all interactions were included (Fig. 5a), but when we calculated multifunctionality as the average of thresholds, this relationship was only marginally significant (Pearson correlation; $r = 0.63$, $p = 0.052$, Fig. 5d). This response was driven by the strong positive relationship between abiotic–biotic coupling and multifunctionality (Fig. 5c–f) as well as by the coupling of all belowground interactions and ecosystem multifunctionality (Fig. 5g–j). In contrast, above–belowground biotic–biotic, belowground biotic–biotic and belowground abiotic–biotic interactions (Fig. 5b–l) were not related to ecosystem multifunctionality. Abiotic–biotic interactions that involved aboveground invertebrates (e.g., interactions between the aboveground invertebrate community and soil properties, Supplementary Table 5) were particularly relevant for ecosystem multifunctionality.

**Invertebrates are important for maintaining ecosystems.** Our results support the view that invertebrates and soil networks are key for multifunctional ecosystems[41] and that more tightly correlated belowground interactions enhance the efficiency of C and N cycling[22]. Our results also suggest that the invertebrate community may promote ecosystem coupling, and thus multifunctionality, in situations where vertebrate mammals have disappeared or have become functionally extinct[1,15]. Yet, not surprisingly, our findings also reveal that systems where all aboveground animals have disappeared are no longer coupled and fall apart (Figs. 2, 3, Supplementary Fig 1 and Supplementary Table 3). Thus, our results show that invertebrates are important for maximising ecological functioning and warn about their on-going decline[4,5,42,43], including pollinators[42,44] and detritivores[45], from ecosystems worldwide, especially in intensively managed agricultural systems where mammals are absent[4,46], but alarmingly also in protected areas[42]. Our results also strongly suggest that maintaining functionally diverse invertebrate communities in grasslands and agro-ecosystems may have positive effects on ecosystem functionality and the provision of ecosystem services[36] through increasing ecosystem coupling. While we undoubtedly need to maintain our efforts in protecting mammals and restoring their ecological roles where needed[47–50], we also need to put considerably more effort into determining and understanding the causes[51] of the progressive and often unseen loss of invertebrates from terrestrial ecosystems[5,42,51] as a way to preserve the complex network of interactions that depends on them.

## Methods

**Study sites.** The experimental exclosure setups were installed within the SNP (IUCN category Ia preserve[31]), in south-eastern Switzerland. The park covers 172 km² of forests and subalpine and alpine grasslands along with scattered rock outcrops and scree slopes. The entire area has been protected from human impact (no hunting, fishing, camping or off-trail hiking) since 1914. Large, fairly homogenous patches of short- and tall-grass vegetation, which originate from different historical management and grazing regimes, cover the park's subalpine grasslands entirely. Short-grass vegetation developed in areas where cattle used to rest (nutrient input) prior to the park's foundation (fourteenth century to 1914)[30,38] and is dominated by lawn grass species such as *Festuca rubra* L., *Briza media* L. and *Agrostis capillaris* L.[30,38]. Today, this vegetation type is intensively grazed by diverse vertebrate and invertebrate communities that inhabit the park and consume up to 60% of the available biomass[24]. Tall-grass vegetation developed where cattle formerly grazed, but did not rest, and is dominated by rather nutrient-poor tussocks of *Carex sempervirens* Vill. and *Nardus stricta* L.[30,38]. This vegetation type receives considerably less grazing, with only roughly 20% of the biomass consumed[24]. Consequently, the two vegetation types together represent a long-term trajectory of changes in grazing regimes. Underlying bedrock of all grasslands is dolomite, which renders these grasslands rather poor in nutrients regardless of former and current land-use regimes.

**Experimental design.** To progressively exclude aboveground vertebrate and invertebrate animals, we established 18 size-selective exclosure setups (nine in short-grass, nine in tall-grass vegetation) distributed over six subalpine grasslands across the SNP[24,25] (Fig. 1). Elevation differences of exclosure locations did not exceed 350 m (between 1975 and 2300 m a.s.l.). The exclosures were established immediately after snowmelt in spring 2009 and were left in place for five consecutive growing seasons (until end of 2013). They were, however, temporarily dismantled every fall (late October after first snowfall) to protect them from avalanches. They were re-established in the same location every spring immediately after snowmelt. Each size-selective exclosure setup consisted of five plots (2 × 3 m) that progressively excluded aboveground vertebrates and invertebrates from large to small (Fig. 1 and Supplementary Table 1). The plots are labelled according to the guilds that had access to them 'L/M/S/I', 'M/S/I', 'S/I', 'I', 'None'; L = large mammals, M = medium mammals, S = small mammals, I = invertebrates, None = no animals had access; Fig. 1, see also Supplementary Table 1). As we only had permission to have the experimental setup in place for five consecutive growing seasons, the experiment had to be completely dismantled in the late fall of 2013 and all material removed from the SNP.

Our exclosure design was aimed at excluding mammalian herbivores, but naturally also excluded the few medium and small mammalian predators, as well as the entire aboveground invertebrate food web (Supplementary Table 1). A total of 26 large to small mammal species can be found in the SNP, but large apex predators are missing (wolf, bear and lynx; Supplementary Table 1). Reptiles, amphibians and birds are scarce to absent in the subalpine grasslands under study. Only two reptile species occur in the park and they are confined to rocky areas that warm up enough for them to survive. One frog species spawns in an isolated pond far from our grasslands. Only three bird species occasionally feed on the subalpine grasslands. Using game cameras (Moultrie 6MP Game Spy I-60 Infra-red Digital Game Camera, Moultrie Feeders, Alabaster, AL, USA), we did observe that the medium- and small-sized mammals (marmot/hares and mice) were not afraid to enter the fences and feed on their designated plots. We never spotted reptiles, amphibians or birds on camera. We distinguished between 59 higher aboveground-dwelling invertebrate taxa that our size-selective exclosures excluded (Supplementary Table 1; see also 'Methods' for aboveground-dwelling invertebrate sampling below).

The 'L/M/S/I' plot (not fenced) was located at least 5 m from the 2.1 m tall and 7 × 9 m large main electrical fence that enclosed the other four plots (Fig. 1). The bottom wire of this fence was mounted at 0.5 m height and was not electrified to enable safe access for medium and small mammals, while fencing out the large ones. Within each main fence, we randomly established four 2 × 3 m plots separated by 1-m-wide walkways from one another and from the main fence line: (1) the 'M/S/I' plots were unfenced, allowing access to all but the large mammals; (2) the 'S/I' plots (10 × 10 cm electrical mesh fence) excluded all medium-sized mammals. Note that the bottom 10 cm of this fence remained non-electrified to enable safe access for small mammals; (3) the 'I' plots (2 × 2 cm metal mesh fence) excluded all mammals. We double-folded the mesh at the bottom 50 cm to reduce the mesh size to smaller than 1 × 1 cm openings; and (4) the 'None' plots were surrounded by a 1 m tall mosquito net (1.5 × 2 mm) to exclude all animals. The top of the plot was covered with a mosquito-meshed wooden frame mounted to the corner posts (roof). We treated these plots a few times with biocompatible insecticide (Clean kill original, Eco Belle GmbH, Waldshut-Tiengen, Germany) to remove insects that might have entered during data collection or that hatched from the soil, but amounts were negligible and did not impact soil moisture conditions within these plots.

To assess whether the design of the 'None' exclosure (mesh and roof) affected the response variables within the plots and, therefore, influenced the results, we established an additional six 'micro-climate control' exclosures (one in each of the six grasslands)[24,25]. These exclosures were built as the 'None' exclosures but were open at the bottom (20 cm) of the 3-m side of the fence facing away from the prevailing wind direction to allow invertebrates to enter. A 20-cm high and 3-m long strip of metal mesh was used to block access to small mammals. Thus, this construction allowed a comparable micro-climate to the 'None' plots, but also a comparable feeding pressure by invertebrates to the 'I' plots. We compared various properties within these exclosures against one another to assess if our construction altered the conditions in the 'None' plots (Supplementary Table 6). We showed that differences in plant (e.g., vegetation height and aboveground biomass) and soil properties (e.g., soil temperature and moisture) found between the 'I' and the 'None' treatments were not due to the construction of the 'None' exclosure, but a function of animal exclusions, although the amount of UV light reaching the plant canopy was significantly reduced[24] (Supplementary Table 6).

**Aboveground invertebrate sampling.** Aboveground invertebrates were sampled with two different methods to capture both ground- and plant-dwelling organisms: (1) we randomly placed two pitfall traps (67 mm in diameter, covered with a roof) filled with 20% propylene glycol in one 1 × 1 m subplot of the 2 × 3 m treatment plots in spring 2013 (May) and emptied them every 2 weeks until late September 2013[40,52]. A pitfall trap consisted of a plastic cylinder (13 cm depth, 6.75 cm diameter). Within each cylinder we placed a 100 ml plastic vial with outer diameter 6.70 cm and on top of the cylinder we placed a plastic funnel to guide the invertebrates into the vials. Each trap was covered with a cone-shaped and transparent

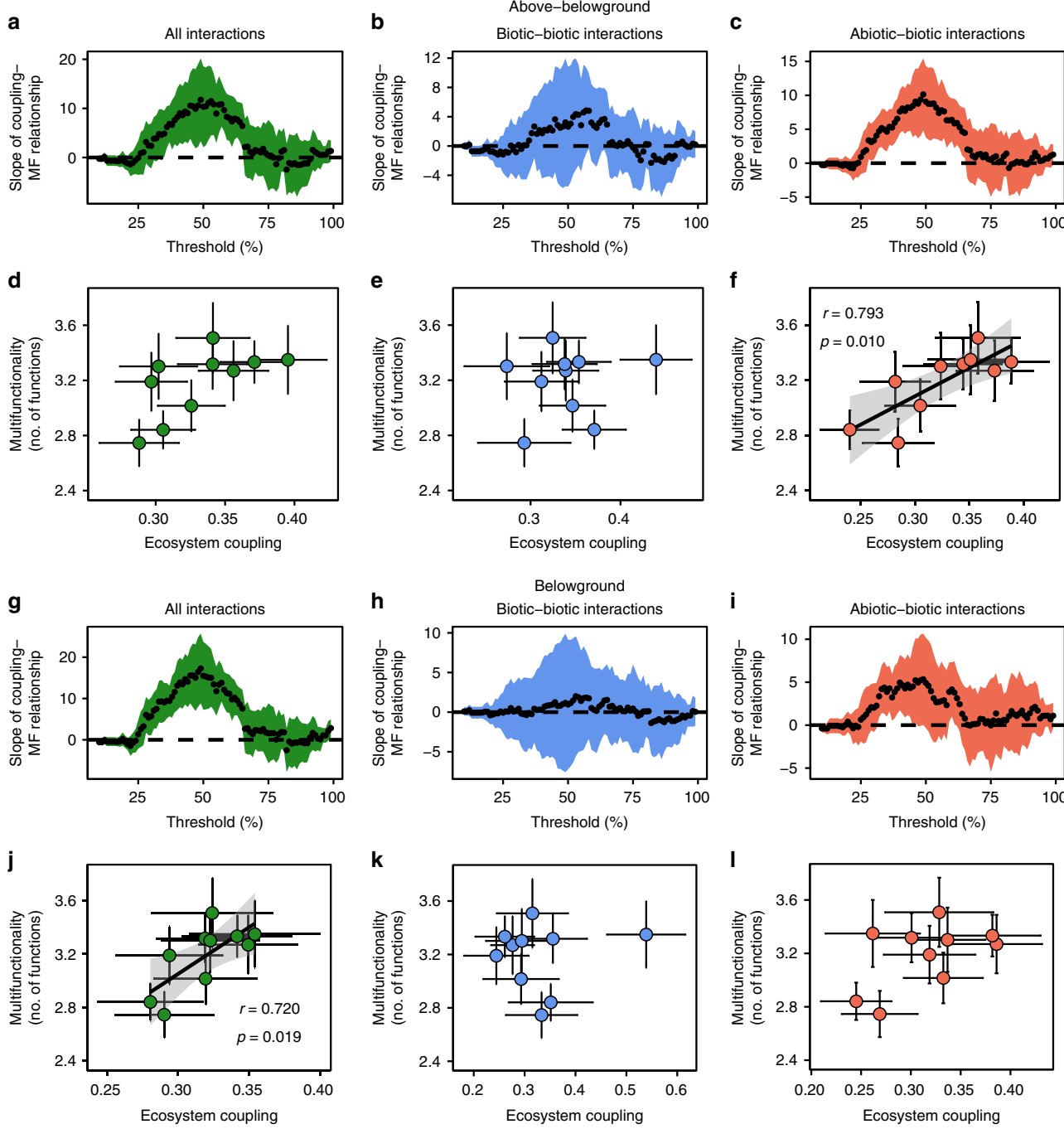

**Fig. 5** Ecosystem coupling–multifunctionality relationship with progressive animal exclusions. Ecosystem coupling was calculated for above–belowground interactions (**a**–**f**) and belowground interactions (**g**–**l**). Ecosystem multifunctionality (MF) was calculated using the multiple threshold approach for thresholds between 10 and 99% (**a**–**c**, **g**–**i**) and for the average of all thresholds intervals (multifunctionality; **d**–**f**, **j**–**l**). The ecosystem coupling–multifunctionality relationship is shown for **a**–**d** all interactions, **b**–**e** biotic–biotic interactions, **c**–**f** abiotic–biotic interactions involving above- and belowground constituents, and **g**, **j** all interactions, **h**–**k** biotic–biotic interactions, and **i**–**l** abiotic–biotic interactions involving belowground constituents only. Bars in **d**–**f** and **j**–**l** represent the standard error of the mean of coupling and multifunctionality within each combination of experimental treatment and vegetation type. $n = 10$. See Supplementary Table 4 for correlations between ecosystem coupling and individual ecosystem functions

plastic roof to protect the trap from rain[40,52]. Note that in the 'None' plots only one trap was placed as control to check for effectiveness of the exclosure. (2) We vacuumed all invertebrates from a 60 × 60 cm area on another 1 × 1 m subplot with a suction sampler (Vortis, Burkhard manufacturing CO, Ltd., Rickmansworth, Hertfordshire, UK) every month from June to September 2013[40,52]. For this purpose, we quickly placed a square plastic frame (60 × 60 × 40 cm) with a closable mosquito mesh sleeve attached to the top edge of the box into the plot from the outside. The suction sampler was then inserted into through the sleeve and operated for 45 s to collect the invertebrates[40,52].

We sorted the ≈100,000 individuals collected with both methods by hand and identified each individual morphologically to the lowest taxonomic-level feasible (59 taxa, including orders, suborders, subfamilies, families; phylum for Mollusca). These taxa belonged to the following feeding types: 19 herbivores, 16 detritivores, 9 predators, 8 mixed feeders, 5 omnivores and 2 non-classified feeders (or not feeding as adults)[52]. We summed the numbers from the two pitfall traps and the suction sampling over the course of the 2013 season to represent the aboveground invertebrate abundance and community composition of a plot. Note: we did not specifically attempt to catch flying invertebrates with, e.g., sticky traps, thus a few

flying insects may have been missed with our vacuum sampling approach.

**Sampling of plant properties**. The vascular plant species composition was assessed at peak biomass every summer (July) by estimating the frequency of occurrence of each species with the pin count method in each plot[53]. A total of 172 taxa occurred within our 90 plots and we calculated plant species richness for each plot separately. We used the 2013 data in this study. Plant quality was assessed every year in July and September; here we use plant quality at the end of the experiment (September 2013). Two 10 × 100-cm-wide strips of vegetation per plot were clipped, combined, dried at 65 °C and ground (Pulverisette 16, Fritsch, Idar-Oberstein, Germany) to pass through a 0.5 mm sieve. Twenty randomly selected samples across all treatments were analysed for N (Leco TruSpec Analyser, Leco, St. Joseph, Michigan, USA)[54]. Nitrogen concentrations of the other samples were then estimated from models established for the experiment and the entire SNP relating Fourier transform-near infra-red reflectance (FT-NIR) spectra to the measured values of N using a multi-purpose FT-NIR spectrometer (Bruker Optics, Fällanden, Switzerland)[54]. Root biomass was sampled every fall by collecting five 2.2 cm diameter × 10 cm deep soil samples (Giddings Machine Company, Windsor, CO, USA) per plot (450 samples per year). The samples were dried at 30 °C and roots were sorted from the sample by hand. We sorted each sample for 1 h, which allowed to retrieve over 90% of all roots present in the samples[24]. The roots were then dried at 65 °C for 48 and weighed to the nearest mg. We averaged the values per plot and used the 2013 data only in this study.

**Sampling of edaphic communities**. In 2009, 2010 and 2011, we collected three composited soil samples (5 cm diameter × 10 cm depth; AMS Samplers, American Falls, ID, USA) and assessed bacterial community structure using T-RFLP profiling[55–57]. We detected a total of 89 operational taxonomic units (OTUs). These values are in accordance with other studies that report OTU richness[58–60] using T-RFLP profiling, a method that detects the most abundant, and thus likely, the most relevant, taxa. We averaged the data over the 3 years of collections for our calculations. Microbial biomass carbon (MBC) was determined with the substrate-induced method[61] every fall (September) between 2009 and 2013 by collecting three mineral soil samples (5 cm diameter × 10 cm mineral soil core, AMS Samplers, American Falls, ID, USA). The three samples were combined (90 samples for each sampling year), immediately put on ice, taken to the laboratory, passed through a 2-mm sieve and stored at 4 °C. Again, we only used the 2013 data in this study.

Soil samples (5 cm diameter × 10 cm depth) to extract soil arthropods were collected in June, July and August 2011 with a soil corer lined with a plastic sleeve to ensure an undisturbed sample (total of 270 samples). The plastic lined core was immediately sealed on both ends using cling film and put into a cooler. All plots were sampled within 3 days and the extraction of arthropods started the evening of the sampling day using a high-gradient Tullgren funnel apparatus[54,62]. Samples were kept in the extractor for 4 days and the soil arthropods were collected in 95% ethanol. All individuals were counted and each individual was identified morphologically to the lowest-level feasible [76 taxa, including orders, suborders, subfamilies, families (Protura, Thysanoptera, Aphidina, Psylina, Coleoptera, Brachycera, Nematocera, Auchenorryncha, Heteroptera, Formicidae); sub-phylum for Myriapoda, for Acari and Collembola we also included morpho-species]. We also included larval stages (nine of the 76 taxa)[54]. All data were summed over the season. A detailed species list for mites and collembolans is published[54]. Earthworms are rare in the SNP and therefore were not included. We collected eight random 2.2 cm diameter × 10 cm deep soil cores from each plot in September 2013 to assess the soil nematode community composition. The samples were mixed and the nematodes were extracted from 100 ml of fresh soil using Oostenbrink elutriators[63]. All nematodes in a 1 ml of the 10 ml extract were counted, a minimum of 150 individuals per sample were identified to genus or family level using ref. [64]. The numbers of all nematodes were extrapolated to the entire sample and expressed for a 100-g dry sample. In total, we identified 63 genus or family levels[54]. A list of all the nematode taxa identified is published[39] (http://www.oikosjournal.org/appendix/oik-03341).

We are aware that sampling soil microbes from 2009 to 2011 and soil arthropods in 2011 was not ideal, but we are positive that this does not bias the results. Most of the parameters measured in our experiment either already showed a treatment response after the first growing season (e.g., plant biomass) or did not respond over the entire time experiment (e.g., microbial biomass C; Supplementary Fig 2). The microbial community composition (2009–2011) was highly influenced by inter-annual differences in temperature and precipitation, but did not differ between treatments or vegetation types[55]. We therefore felt comfortable using the 2009 through 2011 data for describing the soil microbial community in our experimental treatments. Similarly, we are positive that our soil arthropod data are representative. We did assess soil arthropods in August 2012 and found no differences to the August 2011 data. However, we did not feel comfortable combining the 2011 June, July and August data with only August data for 2012 for our analyses.

**Sampling of soil properties**. We collected three soil samples (5 cm diameter × 10 cm depth) in each plot in September 2013 after removing the vegetation. First, we

collected the typically 1–3 cm in deep top layer of mineral soil rich in organic matter (i.e., surface organic layer or rhizosphere) with a soil corer (AMS Samples, American Falls, ID, USA). Second, we collected a 10 cm mineral soil core beneath this surface layer. The cores for each layer were composited, dried at 65 °C for 48 h and fine-ground to pass a 0.5 mm screen. We then analysed all samples for total C using a Leco TruSpec Analyser (Leco, St. Joseph, MI, USA). Mineral soil pH was measured potentiometrically in 1:2 soil:CaCl$_2$ solution with an equilibration time of 30 min.

Soil net N mineralisation was assessed during the 2013 growing season[25]. For this purpose, we randomly collected a 5 cm diameter × 10 cm deep soil sample with a soil corer (AMS Samples, American Falls, ID, USA) after clipping the vegetation in June 2013. After weighing and sieving (4 mm mesh) the soil, we extracted a 20 g subsample in 1 mol l$^{-1}$ KCl for 1.5 h on an end-over-end shaker and thereafter filtered it through ashless-folded filter paper (DF 5895 150, ALBET LabScience, Hahnenmühle FineArt GmbH, Dassel, Germany). From these filtrates, NO$_3^-$ were measured colorimetrically[65] and NH$_4^+$ with flow injection analysis (FIAS 300, PerkinElmer, Waltham, MA, USA)[25]. We dried the rest of the sample 105 °C to constant mass to determine fine-fraction bulk density. A second soil sample was collected within each plot in June 2013 with a corer lined with a 5 × 13 cm aluminium cylinder. The corer was driven 11.5 cm deep into the soil so that the top 1.5 cm of the cylinder remained empty. Into this space we placed a polyester bag (250 μm) filled an ion-exchanger resin to capture the incoming N. The bag was filled with a 1:1 mixture of acidic and alkaline exchanger resin (ion-exchanger I KA/ion exchanger IIIAA, Merck AG, Darmstadt, Germany). We then removed 1.5 cm soil at the bottom of the cylinder and placed a second resin exchanger bag into this space to capture the N leached from the soil column. To assure that the exchange resin was saturated with H$^+$ and Cl$^-$ prior to filling the bags, the mixture was stirred with 1.2 ml l$^{-1}$ HCl for 1 h and then rinsed with demineralised water until the electrical conductivity of the water reached 5 μm cm$^{-1}$. The cylinder with the resin bags in place was reinserted into the soil with the top flush to the soil surface and incubated for 3 months. We recollected the cylinders in September 2013. Each resin bag and 20 g of sieved soil (4 mm mesh) from each cylinder were separately extracted with KCl and NO$_3^-$ and NH$_4^+$ concentrations were measured. Nitrate and NH$_4^+$ concentrations of all samples were then converted to a content basis by multiplying their values with fine-fraction bulk density. Net N mineralisation was thereafter calculated as the difference between the N content of the samples collected at the end of the 3-month incubation (including the N extracted from the bottom resin bag) and the N content at the beginning of the incubation[25].

Soil CO$_2$ emissions were measured every 2 weeks between 0900 and 1700 h from early May through late September 2013 with a PP-Systems SRC-1 soil respiration chamber (15 cm high, 10 cm diameter; closed circuit) attached to a PP-Systems EGM-4 infra-red gas analyser (PP-Systems, Amesbury, MA, USA) on two locations per plot[24]. The chamber was placed on randomly placed, permanently installed PVC collars (10 cm diameter) driven 5 cm into the soil at the beginning of the study[24]. Freshly germinated plants growing within the collars were removed prior to each measurement to avoid measuring plant respiration or photosynthesis. The two measurements collected per plot and sampling date were averaged.

Soil moisture (with time domain reflectometry; Field-Scout TDR-100, Spectrum Technologies, Plainfield, IL, USA) and temperature (with a waterproof digital pocket thermometer; Barnstead International, Dubuque, IA, USA) were measured at five random locations per plot every 2 weeks during the growing seasons during the experiment for the 0–10 cm depth[24,25]. As soil moisture and soil temperature were highly negatively correlated[24], we only used soil moisture for this study. We used plot-level averages of all values available to capture soil moisture variability during the 5 years of the experiment. The results remained unchanged when we only used soil moisture from the 2013 growing season.

**Numeral calculations and statistical analyses**. We conducted principal component analyses (PCAs; unscaled) at the complete data set level using the abundances of each taxonomical entity to describe each of the five different communities used in this study: aboveground-dwelling invertebrates, vascular plants, soil microorganisms, soil arthropods and soil nematodes. We retained the first two components (PCA axis 1 and PCA axis 2) of each analysis as we found them to adequately represent the temporal and spatial variability of our 90-treatment plots in previous studies[55,66,67]. Together, the two components explained a total of 71.70% of the variation for aboveground invertebrates, 44.36% for plants, 44.85% for soil microorganisms, 61.63% for soil arthropods and 77.19% for soil nematodes. In addition, we used soil pH and soil organic C content as a proxy for soil chemical properties, soil bulk density as a proxy for soil physical properties and soil moisture (negatively correlated with soil temperature) as a proxy for soil micro-climatic conditions for an overall total of 14 constituents.

We calculated ecosystem coupling[9] for each exclosure treatment within each vegetation type (i.e., 2 × 5 treatment combinations in total) as an integrated measure of pairwise ecological interactions between ecosystem constituents representing ecological communities and the soil abiotic environment. These ecological interactions are defined by non-parametric Spearman rank correlation analyses between two constituents, excluding interactions involving two abiotic constituents (e.g., soil pH vs. soil moisture) and interactions between the first (PC1) and second (PC2) component of each community type, as these are orthogonal by

definition. Interactions between abiotic constituents were excluded from the analyses because the focus of our study was on communities and how they interact with one another and their surrounding environment; therefore, including abiotic–abiotic interactions was not of interest here. Given that the effectiveness of our experimental design resulted in that no community composition data of aboveground-dwelling invertebrates was available for the 'None' plots (all animals excluded), only 13 instead of 14 constituents were included in the ecosystem coupling calculations for this treatment. The complete absence of aboveground invertebrates represents the most extreme case of disturbance between aboveground animal communities and the rest of the ecosystem constituents. This may have resulted in a slight overestimation of ecosystem coupling for these plots.

Coupling was calculated for each treatment within each vegetation type (i.e., based on nine replicates each), considering a total of 80 interactions (56 in the case of the 'None' treatment) per vegetation type. We considered a total of 40 biotic–biotic interactions (i.e., concerning two community-level principal components such as plants and microbes; 24 in the case of the 'None' treatment) and 40 abiotic–biotic (i.e., concerning one community-level principal component and one abiotic factor, e.g., plant community and soil properties; 32 in the case of the 'None' treatment). To establish whether constituents were significantly and positively coupled within treatments (i.e., the average of their correlation coefficients were greater than in a null model where correlation only happens by chance), we calculated one-tailed $p$ values based on permutation tests with 999 permutations.

We considered six ecosystem functions and process rates commonly used to assess ecosystem functioning[36,68]. Plant N content represents a measure of forage quality, while plant richness has been shown to stabilise biomass production, thus allowing the system to respond to changes in herbivory. Soil net N mineralisation, soil respiration, root biomass and microbial biomass represent fluxes or stocks of energy. For all functions and processes, higher values represent higher functioning[36]. All these variables were measured in the last year of the experiment (2013). We then quantified ecosystem multifunctionality using the multiple threshold approach[36,37], which considers the number of functions that are above a certain threshold, over a series of threshold values (typically 10–99%) that are defined based on the maximum value of each function. We weighted all our functions equally for these calculations[36]. The number of functions in a plot with values higher than a given threshold value for the respective function is summed up. The sum represents ecosystem multifunctionality for that plot. Given that choosing any particular threshold as a measure of ecosystem multifunctionality is arbitrary, we calculated the average of thresholds from 10 to 90% (in 10% intervals) as a more integrated representation of ecosystem multifunctionality.

We used Pearson correlations to explore the relationships between ecosystem coupling (all interactions, biotic–biotic interactions, abiotic–biotic interactions involving above- and belowground constituents, and all interactions, biotic–biotic interactions, abiotic–biotic interactions involving belowground constituents only) and ecosystem multifunctionality by calculating the slopes of all relationships between ecosystem coupling and multifunctionality for all thresholds between 10 and 99%. We also related ecosystem coupling with the average of multifunctionality at thresholds between 30 and 80% and considered this correlation as a robust indication of the type of association between these two variables. In addition, we explored the relationships between ecosystem coupling (all interactions, biotic–biotic interactions, abiotic–biotic interactions involving above- and belowground constituents, and all interactions, biotic–biotic interactions, abiotic–biotic interactions involving belowground constituents only) and individual ecosystem functions. All relationships ($n = 10$) between correlation coefficients (based on nine replicates each) of two individual constituents and ecosystem functions/multifunctionality are shown in Supplementary Tables 4 and 5. The effects of exclosures and vegetation type on individual functions and multifunctionality were evaluated using linear mixed-effects models ('lme' function of the nlme package), with exclosure and vegetation type as fixed effects and fence as a random factor (Fig. 4). All statistical analyses and numerical calculations were done in R version 3.4.0[69].

## Data availability

The data set generated during and/or analysed by the current study will be made available upon acceptance of the paper at the environmental data portal EnviDat (DOI: 10.16904/envidat.44).

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

## Acknowledgements

We would like to thank various employees and volunteers of the Swiss Federal Institute for Forest, Snow and Landscape Research (WSL) and the Swiss National Park for assistance with fence construction and maintenance, vegetation and soil sampling and processing. We thank Alan Haynes, Melanie Hodel, Ursina Raschein and Henk Duyts for their work in plant, microbe, soil arthropod and nematode community assessment, Bigna Stoffel, Vera Baptista, Anna Schweiger, Annatina Zingg and Seraina Cappelli for sorting roots, Roman Alther, Bieke Boden, Monika Carol Resch, Charlotte Schaller, Magdalena Steiner, Silvan Stöckli and Peter Wirz for sampling and sorting aboveground-dwelling invertebrates. We are grateful to Dieter Trummer for developing the prototype of our exclosures. We thank Pablo Hueso for his help with the artwork in Fig. 1. Jennifer Firn, Yann Hautier and Loïc Pellissier provided invaluable comments on earlier versions of this manuscript. This study was funded by the Swiss National Science Foundation, SNF grant no. 31003A_122009/1 and SNF grant no. 31003A_140939/1 to A.C.R. and M.S. R.O.-H. acknowledges funding from a Juan de la Cierva-Incorporación fellowship (IJCI-2014-21252).

## Author contributions

A.C.R. and M.S. developed the experiment and overall research idea. A.C.R coordinated the project. A.C.R., J.K.B. and R.O.-H. developed and framed the research question. R.O.-H analysed the data. M.S., A.C.R., S.Z., B.F., D.S.P.-D., M.D.B., D.J.G. and M.L.V. were in charge or contributed to laboratory analyses, invertebrate or nematode sampling and identification and/or plant data collection. A.C.R., R.O.-H. and W.H.P. wrote the paper with contributions and input of all authors.
