## [Peer Review File · Nature Communications]

Reviewers' comments:

Reviewer #1 (Remarks to the Author):

In this manuscript, Risch and colleagues report on an impressive field study on the exclusion of various size classes of aboveground animals in alpine grassland to study relationships between biotic and abiotic ecosystem variables (called 'ecosystem coupling'). Via the use of different exclosures, Risch and colleagues excluded vertebrates and invertebrates for five consecutive years and performed assessments of multiple ecosystem functions. Furthermore, they calculated a standardized index of ecosystem multifunctionality and performed correlation analyses among different abiotic and biotic variables. The authors find that animals influence ecosystem coupling and multifunctionality and conclude that invertebrates are important for the functioning of ecosystems.

The study by Risch and colleagues presents novel results, put into an interesting ecological context by exploring responses of ecosystem coupling and multifunctionality. This topic is of high interest because many authors have highlighted the need to implement multitrophic interactions in ecological studies on biodiversity change, and recent studies suggest a rapid decline in animals and the processes they drive (e.g. Hallmann et al. 2017). Thus, I think this contribution is timely and has the potential to influence this field of research. Having that said, I also think that the manuscript needs to be revised substantially to present a more balanced and clear message. As I think that the study itself has been performed very well, I will mostly comment on the conceptual frameworks of ecosystem coupling, ecosystem multifunctionality, and on the hypotheses.

Ecosystem coupling. This is an interesting concept, yet a rather simple approach to assess biotic interactions and interactions between biotic and abiotic ecosystem properties. Although papers may have been published using this approach, it needs (at least) some critical discussion in the main text and in the supplementary information from my point of view. Using correlations to assess biotic interactions comes with a suite of potential caveats. For instance, one could easily imagine two opposing mechanisms that contribute to an overall neutral relationship between two focal variables. Relatedly, some biotic interactions like trophic relationships may cause all kinds of correlations, positive, neutral, and negative ones. Some recent works have provided examples how to quantify biotic interactions (e.g., Meyer et al 2015, Roslin et al 2017, Sobral et al 2017), and such examples could be given in a caveats and outlook section.

Usage of ecosystem multifunctionality. Ecosystem multifunctionality is another 'hip' concept that allows to summarize the provisioning of multiple ecosystem functions. However, it also comes with some issues that have not been well reflected in the present form of the manuscript. For instance, a question is which functions enter and how they enter the calculation of multifunctionality. What could be the role of trade-offs among different functions? I think the use of ecosystem multifunctionality (and even more importantly: the single functions entering the calculation) should be better justified. A recently published paper by Manning et al (2018) may help framing this more convincingly. Moreover, Figure 4g suggests that ecosystem multifunctionality is high in treatments without any animals ('None' treatment); a provocative question related to how ecosystem multifunctionality is used here: given the rationale in most parts of the manuscript, wouldn't one conclude that defaunated landscapes (at least aboveground) should provide very high levels of ecosystem multifunctionality? Lines 160-162: I do not understand this conclusion as Figure 4g suggests that any reduction in animals will result in elevated multifunctionality... I think that those questions reflect how difficult the use of multifunctionality can be.

Hypotheses. I doubt that those hypotheses are well prepared. For instance, I was surprised to read about the different strengths of expected effects from the text above. I think some more explanation is required to make the hypotheses a logical consequence of the introduction.

Main figures: mention how positive versus negative correlations are treated?

Methods: last sentence on page 20: information missing in brackets.

Methods: page 23: 89 OTUs seems to be very low.

Methods: page 24: change 2001 to 2011?

References used in the evaluation:

Hallmann et al (2017) More than 75 percent decline over 27 years in total flying insect biomass in protected areas, Plos One, <http://journals.plos.org/plosone/article?id=10.1371/journal.pone.0185809>

Manning et al (2018) Redefining ecosystem multifunctionality. *Nature Ecology & Evolution* 2, 427–436 <https://www.nature.com/articles/s41559-017-0461-7>

Meyer et al (2015) Towards a standardized Rapid Ecosystem Function Assessment (REFA). *TREE* 30(7): 390-7. doi: 10.1016/j.tree.2015.04.006. <https://www.ncbi.nlm.nih.gov/pubmed/25997592>

Roslin, T. et al. *Science* 356, 742-744 (2017).

Sobral, M. et al. *Nature Ecol. Evol.*, 1, 1670–1676 (2017). https://www.nature.com/articles/s41559-017-0334-0?WT.mc_id=COM_NE_coEov_1710_Sobral

Reviewer #2 (Remarks to the Author):

Risch and colleagues present a manuscript relating the results of a five year experiment in which they progressively removed herbivores from an herbaceous system by size-class and examined the consequences for ecosystem connectivity and multi-functionality. While the idea of size-selective removal of herbivores is not particularly novel, the application of this experimental framework to questions of ecosystem coupling and multifunctionality certainly is. The experiment is conceptually interesting and important, and the experimental and statistical methods are sound. In particular, the large number of functions considered when testing for multifunctionality is commendable and beyond most published work. I previously reviewed this manuscript for a different journal, and the most recent draft has addressed the majority of my previous comments. A few issues remain, outlined below.

(1) The set-up of this paper nicely develops the history of herbivore declines globally. However, hypotheses are still put forward related to herbivore effects on ecosystem coupling without any background for why these hypotheses are made. Is there more background that can be included on why different size classes of herbivores might be expected to impact coupling differently?

(2) The ecosystem coupling results could use a little more discussion to place them in context with other work and explain the complex concept.

(3) Why were interactions between abiotic constituents excluded from the models?

Reviewer #3 (Remarks to the Author):

Using a series of enclosure experiments, the authors explore how changing species interactions alter the coupling of biotic-biotic and abiotic-biotic relationships. They predicted that losing large animals would have a larger (and negative) impact on function. Subsequent removal of smaller taxa would have less of an impact on function. The experiment is, relative to other ecological experiments, novel and long-term and the results are interesting.

This is a neat paper and a novel way to think about how shifting interactions may cascade to impact ecosystems and the services they provide. I'll be interested to see if they can dig into the mechanisms in future papers. I think the framing of the paper is a bit oversold given what little information we

have on species interactions and their potential losses as well as the impact those losses will have on ecosystem function (a fact highlighted by the authors), but it's still a novel and cool experimental study with some interesting results that will engage the ecological and scientific community. The authors could tone down their proclamations a bit.

It took me a few reads and a deep dive into the methods to figure out the experimental design. I suggest that the authors rework Figure 1. As it stands it is hard to determine the experimental design (how many sites, what's the sample size, etc.). I think explicitly clarifying the design in Figure 1 would make it easier for readers to follow what was manipulated and how. I also think the L/M/S/I labeling is confusing and unnecessary, can you just spell it out (or use a picture of some sort) — but these are easy things to fix.

Title: What is a critical function — or an uncritical function? I think you could omit "critical"

I don't really understand how they calculated coupling using the Disruption Index — I think someone more familiar with these sorts of statistical methods should comment on their use. I was curious about the sample size and the correlation — a sample size of 9 seems small to use in this sort of analyses (Figure 3), but I recognize how much work this study must have been.

One thing that stuck out to me was the difference between Figure 2 and Figure 3. The pattern in Figure 3 is striking — as you move from left to right, however the patterns in figure 2 aren't that different from the null. What's driving the difference between the figures?

Line 143 — Why are interactions in soils, from a 5 large homogenized soil core, larger than variation in above ground communities? I imagine plants, insects, and mammals would also be patchy in a meadow?

Lines 162-163 — this statement seems like an overreach to me. Increasingly common? I don't think it's a common pattern (not yet anyway). Please tone it down.

Line 170 — There are many things that conservation and management programs should do. If you are going to suggest that conservation and management programs should "consider coupling" — you need to say how they should do this. If you can't say how, then leave this out. Also, you need to show it has an impact across different ecosystems for your study highlights variability within the same type of system. This is a neat study and a very cool dataset, but don't extend the ecology beyond a few meadows in Switzerland without having something big behind it that allows you to extrapolate — like modeling output or experiments across many system types.

Reviewers' comments:

Reviewer #1 (Remarks to the Author):

In this manuscript, Risch and colleagues report on an impressive field study on the exclusion of various size classes of aboveground animals in alpine grassland to study relationships between biotic and abiotic ecosystem variables (called 'ecosystem coupling'). Via the use of different exclosures, Risch and colleagues excluded vertebrates and invertebrates for five consecutive years and performed assessments of multiple ecosystem functions. Furthermore, they calculated a standardized index of ecosystem multifunctionality and performed correlation analyses among different abiotic and biotic variables. The authors find that animals influence ecosystem coupling and multifunctionality and conclude that invertebrates are important for the functioning of ecosystems.

The study by Risch and colleagues presents novel results, put into an interesting ecological context by exploring responses of ecosystem coupling and multifunctionality. This topic is of high interest because many authors have highlighted the need to implement multitrophic interactions in ecological studies on biodiversity change, and recent studies suggest a rapid decline in animals and the processes they drive (e.g. Hallmann et al. 2017). Thus, I think this contribution is timely and has the potential to influence this field of research. Having that said, I also think that the manuscript needs to be revised substantially to present a more balanced and clear message. As I think that the study itself has been performed very well, I will mostly comment on the conceptual frameworks of ecosystem coupling, ecosystem multifunctionality, and on the hypotheses.

Dear reviewer,

Many thanks for your kind and very insightful comments about our study, which have greatly helped to produce a clearer manuscript. This is especially true for comments about key theoretical concepts such as ecosystem coupling and multifunctionality, which were evidently not as clearly defined as we initially thought. We have now also done our best to present clearer hypotheses solidly grounded in the existing literature.

Ecosystem coupling. This is an interesting concept, yet a rather simple approach to assess biotic interactions and interactions between biotic and abiotic ecosystem properties. Although papers may have been published using this approach, it needs (at least) some critical discussion in the main text and in the supplementary information from my point of view. Using correlations to assess biotic interactions comes with a suite of potential caveats. For instance, one could easily imagine two opposing mechanisms that contribute to an overall neutral relationship between two focal variables. Relatedly, some biotic interactions like trophic relationships may cause all kinds of correlations, positive, neutral, and negative ones. Some recent works have provided examples how to quantify biotic interactions (e.g., Meyer et al 2015, Roslin et al 2017, Sobral et al 2017), and such examples could be given in a caveats and outlook section.

We agree with you in that, despite the attractiveness of the ecosystem coupling concept as an integrative, whole-system indicator of trophic and non-trophic interactions, its apparent simplicity does not come without caveats. As you point out, many other papers have been published using Spearman rank correlations, including papers strictly focused on ecological networks. Therefore, we are now critically discussing why this concept is appropriate for our study, even if we cannot capture the separate role of opposing ecological mechanisms in a simple index. This is because, in our study, ecosystem coupling is calculated for ecological communities instead of individual species. Thus, our biotic constituents are based on the composition of multispecies communities represented by principal components. As such the opposing mechanisms are already included in the axis scores and ecosystem coupling can be calculated as the average strength of all interactions, while the "sign" of the relationship does not longer represent "positive" or "negative" interactions between individual species. We embedded this information in the first paragraph (lines 50-64) where we introduce the concept of ecosystem coupling.

Usage of ecosystem multifunctionality. Ecosystem multifunctionality is another ‘hip’ concept that allows to summarize the provisioning of multiple ecosystem functions. However, it also comes with some issues that have not been well reflected in the present form of the manuscript. For instance, a question is which functions enter and how they enter the calculation of multifunctionality. What could be the role of trade-offs among different functions? I think the use of ecosystem multifunctionality (and even more importantly: the single functions entering the calculation) should be better justified. A recently published paper by Manning et al (2018) may help framing this more convincingly. Moreover, Figure 4g suggests that ecosystem multifunctionality is high in treatments without any animals (‘None’ treatment); a provocative question related to how ecosystem multifunctionality is used here: given the rationale in most parts of the manuscript, wouldn’t one conclude that defaunated landscapes (at least aboveground) should provide very high levels of ecosystem multifunctionality? Lines 160-162: I do not understand this conclusion as Figure 4g suggests that any reduction in animals will result in elevated multifunctionality... I think that those questions reflect how difficult the use of multifunctionality can be.

We agree with you that a better justification of our variable selection and calculation procedures for ecosystem multifunctionality was needed. Based on your advice, we also recalculated ecosystem multifunctionality using the multiple threshold approach suggested by Manning et al. (2018). However, instead of selecting any threshold as the focus of our analysis (e.g., 50%), we calculated the average of all thresholds between 10-90% (in 10% intervals) and used this value as our multifunctionality metric (lines 133-136; 676-688). To ease the interpretation of these new calculations, we calculated the relationships between the previously used multifunctionality index (based on the average of z-scores) and individual thresholds (10% to 90%) as well as average threshold for this response letter (see Fig R1 below). We found that the average of z-scores and the average threshold values are highly correlated which, in our opinion, not only supports but also strengthens our previous conclusions.

With regard to variable selection, we decided to exclude plant biomass from the calculations. Although it is considered as a function in many studies, including Manning et al. (2018). in our case it is a reflection of lower consumption rates due the loss of animals rather than a reflection of greater productivity. Instead, we included plant richness, a proxy for plant biodiversity, as a function into our calculations.

We also added an explanation about why ecosystem multifunctionality is somewhat lower in the treatment with access to all animals compared to the other treatments with animals present (lines 180-185): “...lower average functionality might reflect the strong top-down control that these animals can have in the absence of large predators (e.g., through biomass consumption and physical disturbance), as found in our system where wolves, bears and lynx were locally extinct by the late 19th century and remain functionally absent”.

Manning, P., van der Plas, F., Soliveres, S., Allan, E., Maestre, F.T., Mace, G., Whittingham, M.J., Fisher, M. Redefining ecosystem multifunctionality. *Nature Ecology and Evolution* 2: 427-436

Fig R1. Individual thresholds for multifunctionality and (lowest panel) average threshold plotted against the average of z-scores.

Hypotheses. I doubt that those hypotheses are well prepared. For instance, I was surprised to read about the different strengths of expected effects from the text above. I think some more explanation is required to make the hypotheses a logical consequence of the introduction.

Critically re-reading our manuscript in light of your comments, we agree in that the presentation of our hypotheses could be improved. Accordingly, in this revised version of our manuscript we present our expectations in a way that we hope is much clearer to the reader and also solidly grounded in the available literature. In this sense, we are now hypothesizing that “Large mammals are often assumed to drive trophic interactions via top-down effects and impact other communities and abiotic properties, for example, via predation, grazing, resource competition or facilitation. We therefore predicted that the loss of large animals would reduce ecosystem coupling, i.e., weaken correlations among constituents, more than a subsequent loss of smaller animals. Expecting that stronger links between communities and also between communities and their environment result in a more efficient flow of energy and matter through the ecosystem, we also predicted a positive relationship between ecosystem coupling and functionality..” (lines 84-91).

Main figures: mention how positive versus negative correlations are treated?

As pointed out before in our response to your first comment, our biotic constituents are based on the composition of multispecies communities represented by principal components. As such, opposing ecological mechanisms defining the structure and composition of the communities are already included in the axes scores. Thus the “sign” of the relationship between component is not relevant here and does not represent “positive” or “negative” interactions. To clarify this aspect, we have now strengthened our explanation of how ecosystem coupling is calculated in the main text (lines 50-64). We are now hoping to make clear why the sign of the correlation does not matter when calculating ecosystem coupling.

Methods: last sentence on page 20: information missing in brackets.

We changed the sentences so that it is clear that we refer to a publication here.

Methods: page 23: 89 OTUs seems to be very low.

We have now added literature showing that the number of OTUs found is within the normal range reported in other studies using this methodology (T-RFLP). We also added some text explaining that T-RFLP is a method that detects the most abundant and thus likely the most relevant taxa. Lines 610-613

Methods: page 24: change 2001 to 2011?

This typo has now been fixed.

References used in the evaluation:

Hallmann et al (2017) More than 75 percent decline over 27 years in total flying insect biomass in protected areas, Plos One,

<http://journals.plos.org/plosone/article?id=10.1371/journal.pone.0185809>

Manning et al (2018) Redefining ecosystem multifunctionality. Nature Ecology & Evolution 2, 427–436 <https://www.nature.com/articles/s41559-017-0461-7>

Meyer et al (2015) Towards a standardized Rapid Ecosystem Function Assessment (REFA). TREE 30(7): 390-7. doi: 10.1016/j.tree.2015.04.006. <https://www.ncbi.nlm.nih.gov/pubmed/25997592>

Roslin, T. et al. Science 356, 742-744 (2017).

Sobral, M. et al. Nature Ecol. Evol., 1, 1670–1676 (2017). https://www.nature.com/articles/s41559-017-0334-0?WT.mc_id=COM_NEcoEov_1710_Sobral

Many thanks for suggesting these relevant references, most of which have now been incorporated into the manuscript. Please see revised reference list.

Reviewer #2 (Remarks to the Author):

Risch and colleagues present a manuscript relating the results of a five-year experiment in which they progressively removed herbivores from an herbaceous system by size-class and examined the consequences for ecosystem connectivity and multi-functionality. While the idea of size-selective removal of herbivores is not particularly novel, the application of this experimental framework to questions of ecosystem coupling and multifunctionality certainly is. The experiment is conceptually interesting and important, and the experimental and statistical methods are sound. In particular, the large number of functions considered when testing for multifunctionality is commendable and beyond most published work. I previously reviewed this manuscript for a different journal, and the most recent draft has addressed the majority of my previous comments. A few issues remain, outlined below.

Dear reviewer,

Thank you very much for your positive re-assessment of our manuscript and for your time dedicated to it. We have again done our best to address your remaining questions and concerns in this revised version of our manuscript. Please find detailed responses to your comments below.

The set-up of this paper nicely develops the history of herbivore declines globally. However, hypotheses are still put forward related to herbivore effects on ecosystem coupling without any background for why these hypotheses are made. Is there more background that can be included on why different size classes of herbivores might be expected to impact coupling differently?

In agreement with your and reviewer #1's comments we have further clarified our expectations and also solidly grounded them in the available literature. In this sense, we are now hypothesizing that given that "Large mammals are often assumed to drive trophic interactions via top-down effects and impact other communities and abiotic properties, for example, via predation, grazing, resource competition or facilitation². We therefore predicted that the loss of large animals would reduce ecosystem coupling, i.e., weaken correlations among constituents, more than a subsequent loss of smaller animals. Expecting that stronger links between communities and also between communities and their environment result in a more efficient flow of energy and matter through the ecosystem, we also predicted a positive relationship between ecosystem coupling and functionality.." (lines 84-91).

The ecosystem coupling results could use a little more discussion to place them in context with other work and explain the complex concept.

Also in agreement with your and Reviewer #1's comments we clarify the concept of ecosystem coupling and its implications. Together with our initial explanation of ecosystem coupling "as the overall strength of correlation-based associations between above- and belowground plant, animal, and microbial communities, and of these communities with their surrounding physicochemical environment" (lines 52-54), we are now further defining this concept by referring to "Visually and analytically, ecosystem coupling can be represented as a "web" or "network" in which individual species are substituted with multispecies communities (e.g., microbes, plants, soil arthropods, nematodes)." (lines 54-57) and further clarify "Under undisturbed conditions we would expect that the communities are more strongly connected with one another and their abiotic environment than under disturbed conditions. The connections can result from both direct and indirect, positive or negative interactions, depending on the communities and the dominant mechanisms involved (e.g., predation, parasitism, symbiosis, facilitation, competition, etc.). A greater number of significant correlations, regardless of their sign, represents greater ecosystem coupling"(lines 57-62).

Why were interactions between abiotic constituents excluded from the models?

We agree with you in that this is a relevant question that was not properly addressed in previous versions of our manuscript. For this reason, we are now including a justification of this decision both in the material and methods section (lines 676-679) and in the main text (lines 131-133). Briefly explained, “interactions between abiotic constituents were excluded from the analyses because the focus of our study was on communities and how they interact with one another and their surrounding environment; therefore, including abiotic-abiotic interactions would not have contributed to our study”.

Reviewer #3 (Remarks to the Author):

Using a series of enclosure experiments, the authors explore how changing species interactions alter the coupling of biotic-biotic and abiotic-biotic relationships. They predicted that losing large animals would have a larger (and negative) impact on function. Subsequent removal of smaller taxa would have less of an impact on function. The experiment is, relative to other ecological experiments, novel and long-term and the results are interesting.

This is a neat paper and a novel way to think about how shifting interactions may cascade to impact ecosystems and the services they provide. I'll be interested to see if they can dig into the mechanisms in future papers. I think the framing of the paper is a bit oversold given what little information we have on species interactions and their potential losses as well as the impact those losses will have on ecosystem function (a fact highlighted by the authors), but it's still a novel and cool experimental study with some interesting results that will engage the ecological and scientific community. The authors could tone down their proclamations a bit.

Dear reviewer,

Many thanks for your insightful comments about our study, which have particularly helped us to narrow down the framing of our manuscript. In this revised version of our article we are now paying special attention to not overselling our results by toning down some of our proclamations, particularly in the conclusions paragraph.

It took me a few reads and a deep dive into the methods to figure out the experimental design. I suggest that the authors rework Figure 1. As it stands it is hard to determine the experimental design (how many sites, what's the sample size, etc.). I think explicitly clarifying the design in Figure 1 would make it easier for readers to follow what was manipulated and how. I also think the L/M/S/I labeling is confusing and unnecessary, can you just spell it out (or use a picture of some sort) — but these are easy things to fix.

We have now re-organized and reworked Figure 1, which now shows the treatments explicitly, the two grassland types and the number of enclosures. We added icons of animals to show what our abbreviations refer to. For clarity and consistency reasons we kept the “L/M/S/I” etc labels in Fig 2 through 5 as there is either no space (Fig 2,4) for the graphical design or in our opinion would make the figure too busy (Fig 3). However, we certainly could change them if the editor thinks that having the drawings would make our manuscript more accessible to our readers. We are also happy to make additional changes to Fig 1 if the editor feels that more clarity is needed or the figure could be further enhanced to help understand our design.

Title: What is a critical function — or an uncritical function? I think you could omit “critical”
The word “critical” has now been omitted from the title as suggested.

I don't really understand how they calculated coupling using the Disruption Index — I think someone more familiar with these sorts of statistical methods should comment on their use.

We acknowledge that, despite their similarities, referring to the Disruption Index in the Material and Methods when talking about ecosystem coupling may have created some confusion. This is the reason why we now avoid mentioning this other index. Instead, we directly explain what ecosystem coupling is in the main text (lines 50-62) and also provide the equation used to calculate it in the Material and Methods section (lines 663ff).

I was curious about the samples size and the correlation — a sample size of 9 seems small to use in this sort of analyses (Figure 3), but I recognize how much work this study must have been.

As you point out, our ecosystem coupling metric should ideally have been based on interactions calculated from more than nine points. But, as you also point out, these calculations are based on a huge dataset from ninety experimental plots concerning five experimental treatments in two vegetation types. Moreover, we estimated coupling based on a total of 80 different interaction types which, in our opinion, make our results solid. To further overcome the limitations imposed by the low number of replicates per treatment, we used Spearman rank correlations instead of Pearson correlations to calculate the average strength of interactions among all our biotic and abiotic ecosystem components (ecosystem coupling).

One thing that stuck out to me was the difference between Figure 2 and Figure 3. The pattern in Figure 3 is striking — as you move from left to right, however the patterns in figure 2 aren't that different from the null. What's driving the difference between the figures?

In Figure 2 all interactions are shown (individual points in the background). However, in Figure 3 we only drew those interactions that represented correlations with p-values smaller than 0.1. Therefore, we think that the apparent disagreement between these two figures is due to the different criteria used (i.e., plotting all interactions in Figure 2 vs. plotting significant interactions only in Figure 3). The use of these two criteria is justified by the different purpose of each figure.

Line 143 — Why are interactions in soils, from a 5-large homogenized soil core, larger than variation in above ground communities? I imagine plants, insects, and mammals would also be patchy in a meadow?

We completely agree with your comment and, for this reason, acknowledge that this result may have seemed contradictory at first sight. However, we think that this result simply represents an unavoidable artifact of our data “due to the smaller number of belowground-only links compared to overall interactions” and that, therefore, is “due to the smaller number of belowground-only links compared to overall interactions and not a reflection of the underlying biology of the system”. (lines 168-169)

Lines 162-163 — this statement seems like an overreach to me. Increasingly common? I don't think it's a common pattern (not yet anyway). Please tone it down

The currently available literature about this topic is contradictory, but we agree with you in that this might appear as an overreach. We have now toned the whole paragraph down by deleting that entire sentence.

Line 170 — There are many things that conservation and management programs should do. If you are going to suggest that conservation and management programs should “consider coupling” — you need to say how they should do this. If you can't say how, then leave this out. Also, you need to show it has an impact across different ecosystems for own study highlights variability within the same type of system. This is a neat study and a very cool dataset, but don't extend the ecology beyond a few

meadows in Switzerland without having something big behind it that allows you to extrapolate— like modeling output or experiments across many system types.

In agreement with your comment, we deleted the statement. We also re-worded this paragraph to emphasize the importance of protecting invertebrates. Nevertheless, our study represents one of the first to show how such a disappearance of invertebrates affects the functioning of the system, so that we think that our local study in Switzerland may serve as a basis for urgently needed investigations about how the human-induced reduction in invertebrates affects ecosystem worldwide. Lines 211-230

REVIEWERS' COMMENTS:

Reviewer #1 (Remarks to the Author):

The authors have done a good job addressing my previous comments. I believe that this could be a very nice contribution to the field of multitrophic biodiversity-ecosystem function research and may be interesting to the broad readership of Nature Communications. However, one important aspect still needs improvement from my point of view: the authors have misinterpreted my suggestion to refer to the Manning et al. (2018) paper. My point was that this paper may help with developing a more convincing conceptual framework that may help to better justify decisions regarding which functions to include in the multifunctionality index and which not. The authors now have just added another calculation (multiple threshold for multifunctionality), but did not better justify which functions they included. They now even removed aboveground plant biomass from the calculation without a solid explanation in the main text.

Reviewer #3 (Remarks to the Author):

The authors handed all my quires and concerns.

REVIEWERS' COMMENTS:

Reviewer #1 (Remarks to the Author):

The authors have done a good job addressing my previous comments. I believe that this could be a very nice contribution to the field of multitrophic biodiversity-ecosystem function research and may be interesting to the broad readership of Nature Communications. However, one important aspect still needs improvement from my point of view: the authors have misinterpreted my suggestion to refer to the Manning et al. (2018) paper. My point was that this paper may help with developing a more convincing conceptual framework that may help to better justify decisions regarding which functions to include in the multifunctionality index and which not. The authors now have just added another calculation (multiple threshold for multifunctionality), but did not better justify which functions they included. They now even removed aboveground plant biomass from the calculation without a solid explanation in the main text.

- We have now added a more detailed explanation about why we included our six ecosystem functions and process rates according to the suggestions by the reviewer and Manning et al. (2018) to the method section (lines 805-831 in pdf of track changes main document). We also added a short statement to the introduction that our functions represent “ecosystem functions and process rates” (line 162 in pdf of track-changes main document).

Based on recommendations from the Manning et al. (2018) paper, we conducted a cluster analysis of our functions that could be used for weighing the functions when calculating multifunctionality (see Fig R1 below). However, this method did not reveal clear main clusters in our dataset and the two clusters found grouped very different types of functions that are not related. For this reason, we decided to keep our calculations with all functions weighed the same.

We decided to omit an explanation (main text and methods) of why we did not include certain variables as functions as this may confuse the reader. In a large and long-term experiment many different things are measured and, in the end, not all are used in a synthesis paper such as the current manuscript. Thus, describing all the potentially available variables and how they were measured to then only mention that they were not used as function or process rate would have removed the focus from our main message.

Finally, although our dataset could certainly contribute to develop a more convincing conceptual framework that may help to better justify future decisions regarding which functions to include in a multifunctionality index, we think that this is beyond the goal of this study. For this reason, we decided to solidly justify the inclusion of the functions considered, taking into account the framework proposed by Manning et al. (2018), but not going any further.

Fig R1. Cluster analyses of ecosystem functions according to Manning et al. (2018). Note that no clear main clusters were found.

Reviewer #1's comments to the author:

I do not have any further comments. Nice work.